# Radiomics Machine Learning Analysis of Clear Cell Renal Cell Carcinoma for Tumour Grade Prediction Based on Intra-Tumoural Sub-Region Heterogeneity

**DOI:** 10.3390/cancers16081454

**Published:** 2024-04-10

**Authors:** Abeer J. Alhussaini, J. Douglas Steele, Adel Jawli, Ghulam Nabi

**Affiliations:** 1Division of Imaging Sciences and Technology, School of Medicine, Ninewells Hospital, University of Dundee, Dundee DD1 9SY, UK; 2Department of Clinical Radiology, Al-Amiri Hospital, Ministry of Health, Sulaibikhat 1300, Kuwait; 3Department of Clinical Radiology, Sheikh Jaber Al-Ahmad Al-Sabah Hospital, Ministry of Health, Sulaibikhat 1300, Kuwait

**Keywords:** clear cell renal cell carcinoma, renal masses, biopsy, computed tomography, radiomics, machine learning, tumour sub-regions, tumour heterogeneity, precision medicine

## Abstract

**Simple Summary:**

Clear cell renal cell carcinoma (ccRCC) accounts for at least 80% of renal tumours worldwide. The grading of clear cell carcinoma is crucial for its management; therefore, it is important to distinguish the ccRCC grade pre-operatively. The aim of this research is to differentiate high- from low-grade ccRCC non-invasively using machine learning (ML) and radiomics features extracted from pre-operative computed tomography (CT) scans, taking into consideration the tumour sub-region that offers the greatest accuracy when grading. Furthermore, radiomics and machine learning were compared with biopsy-determined grading in a sub-group with resection histopathology as a reference standard.

**Abstract:**

Background: Renal cancers are among the top ten causes of cancer-specific mortality, of which the ccRCC subtype is responsible for most cases. The grading of ccRCC is important in determining tumour aggressiveness and clinical management. Objectives: The objectives of this research were to predict the WHO/ISUP grade of ccRCC pre-operatively and characterise the heterogeneity of tumour sub-regions using radiomics and ML models, including comparison with pre-operative biopsy-determined grading in a sub-group. Methods: Data were obtained from multiple institutions across two countries, including 391 patients with pathologically proven ccRCC. For analysis, the data were separated into four cohorts. Cohorts 1 and 2 included data from the respective institutions from the two countries, cohort 3 was the combined data from both cohort 1 and 2, and cohort 4 was a subset of cohort 1, for which both the biopsy and subsequent histology from resection (partial or total nephrectomy) were available. 3D image segmentation was carried out to derive a voxel of interest (VOI) mask. Radiomics features were then extracted from the contrast-enhanced images, and the data were normalised. The Pearson correlation coefficient and the XGBoost model were used to reduce the dimensionality of the features. Thereafter, 11 ML algorithms were implemented for the purpose of predicting the ccRCC grade and characterising the heterogeneity of sub-regions in the tumours. Results: For cohort 1, the 50% tumour core and 25% tumour periphery exhibited the best performance, with an average AUC of 77.9% and 78.6%, respectively. The 50% tumour core presented the highest performance in cohorts 2 and 3, with average AUC values of 87.6% and 76.9%, respectively. With the 25% periphery, cohort 4 showed AUC values of 95.0% and 80.0% for grade prediction when using internal and external validation, respectively, while biopsy histology had an AUC of 31.0% for the classification with the final grade of resection histology as a reference standard. The CatBoost classifier was the best for each of the four cohorts with an average AUC of 80.0%, 86.5%, 77.0% and 90.3% for cohorts 1, 2, 3 and 4 respectively. Conclusions: Radiomics signatures combined with ML have the potential to predict the WHO/ISUP grade of ccRCC with superior performance, when compared to pre-operative biopsy. Moreover, tumour sub-regions contain useful information that should be analysed independently when determining the tumour grade. Therefore, it is possible to distinguish the grade of ccRCC pre-operatively to improve patient care and management.

## 1. Introduction

The grading of RCC has been acknowledged as a prognostic marker for close to a century [1]. The tumour grade provides some insight into how cancer may act. It identifies whether cancer cells are regular or aberrant under a microscope. The more aberrant the cells seem and the higher the grade, the quicker the tumour is likely to spread and expand. Many different grading schemes have been proposed, initially focused on a collection of cytological characteristics and more recently on nuclear morphology. Nuclear size (area, major axis, and perimeter), nuclear shape (shape factor and nuclear compactness), and nucleolar prominence characteristics are the main emphasis in the Fuhrman grading of renal cell carcinomas. Even though Fuhrman grading has been widely used in clinical investigations, its predictive value and reliability are up for discussion. Fuhrman et al. [2] showed, in 1982, that tumours of grades 1, 2, 3, and 4 presented considerably differing metastatic rates. When grade 2 and 3 tumours were pooled into a single cohort, they likewise demonstrated a strong correlation between tumour grade and survival [2]. The International Society of Urologic Pathologists (ISUP) suggested a revised grading system for RCC, commonly known as the WHO/ISUP grading system, in 2012 to address the shortcomings of the Fuhrman grading scheme [3]. This system is primarily based on the assessment of nucleoli and has been approved for papillary and ccRCC tumours [3]. As a result of the recommendation by the World Health Organization (WHO), the system is currently applied internationally [4].

Grading ultimately facilitates the optimal management and treatment of tumours according to their prognostic behaviour concerning their respective grades. For instance, in elderly or very sick patients who have small renal tumours (<4 cm) and high mortality rates, cryoablation, active surveillance, or radiofrequency ablation may be considered to manage their conditions. It is crucial to note that a confident radiological diagnosis of low-grade tumours in active surveillance can significantly impact clinical decisions, hence eliminating the risk of over-treatment. As ccRCC is the most prevalent subtype (8 in 10 RCCs) with the highest potential for metastasis, it requires careful characterisation [5]. High-grade cancers have a poorer prognosis, are more aggressive, have a high risk of post-operative recurrence, and may metastasise. Therefore, it is very important to differentiate between different grades of ccRCC, as high-grade ccRCC requires immediate and exact management. Precision medicine together with personalised treatment has advanced with the advent of cutting-edge technology; hence, clinicians are interested in determining the grade of ccRCC before surgery or treatment, enabling them to better advise patients regarding therapy and even predict cancer-free survival if surgery has been conducted.

The diagnosis of ccRCC grades is commonly carried out based on pre- and post-operative methods. One such pre-operative method is biopsy. However, the accuracy of a biopsy can be influenced by several factors, including the size and location of the tumour, the experience of the pathologist performing the biopsy, and the quality of the biopsy sample [6]. Due to sampling errors, a biopsy may not always provide an accurate representation of the overall tumour grade [7]. Inter-observer variability can also lead to inconsistencies in the grading process. This can be especially problematic for tumours that are borderline between two grades [8]. In some cases, a biopsy may not provide a definitive diagnosis as it only considers the cross-sectional area of the tumour and, compounded with the fact that ccRCC presents high spatial and temporal heterogeneity [9], it may not be representative of the entire tumour [10]. A biopsy also has a small chance of haemorrhage (3.5%) and a rare risk of track seeding (1:10,000) [11]. Due to the limitations highlighted for biopsies [12], radical or partial nephrectomy treatment specimens are usually used as definitive post-operative diagnostic tools for tumour grading. With partial or radical nephrectomy being the definitive therapeutic approach, a small but significant number of patients are subjected to unnecessary surgery, even though their management may not require surgical resection. Nephrectomy also increases the possibility of contracting chronic renal diseases that may result in cardiovascular ailments. Therefore, the precise grading of ccRCC through non-invasive methods is imperative in order to improve the effectiveness and targeted management of tumours.

The assumption in most research and clinical practice is that solid renal masses are homogenous in nature or, if heterogeneous, that they have the same distribution throughout the tumour volume [13]. Recent studies [14] have highlighted that, in some histopathologic classifications, different tumour sub-regions may have different rates of aggressiveness; hence, heterogeneity plays a significant role in tumour progression. Ignoring such intra-tumoural differences may lead to inaccurate diagnosis, treatment, and prognosis. The biological makeup of tumours is complex and, therefore, leads to spatial differences within their structures. These variations may be due to the expression of genes or the microscopic structure [15]. Such differences can be caused by several factors, including hypoxia (i.e., the loss of oxygen in the cells) and necrosis (i.e., the death of cells). This is mostly synonymous with the tumour core. Likewise, high cell growth and tumour-infiltrating cells are factors associated with the periphery [16].

Medical imaging analysis has been shown to be capable of detecting and quantifying the level of heterogeneity in tumours [17,18,19]. This ability enables tumours to be categorised into different sub-regions depending on their level of heterogeneity. In relation to tumour grading, intra-tumoural heterogeneity may prove useful in determining the sub-region of the tumour containing the most prominent features that enable successful grading of the tumour. Radiomics, which is the extraction of high-throughput features from medical images, is a modern technique that has been used in medicine to extract features that would not be otherwise visible to the naked eye alone [20]. It was first proposed by Lambin et al. [21] in 2012 to extract features, taking into consideration the differences in solid masses. Radiomics eliminates the subjectivity in the extraction of tumour features from medical images, functioning as an objective virtual biopsy. A significant number of studies have applied radiomics approaches for the classification of tumour subtypes, grading, and even the staging of tumours [22,23].

Artificial intelligence (AI) is a wide area whose aim is to build machines which simulate human cognitive abilities. It has enabled a shift from human systems to machine systems trained by computers using features obtained from the input data. In recent years, with the advent of AI, there has been tremendous progress in the field of medical imaging. Machine learning, which is a branch of AI, has been used to extract high-dimensional features from medical images and has shown significant ability to perform image segmentation, recognition, reconstruction, and classification. It has also made it easy to quantify and standardise medical image features, thereby acting as an intermediary between clinics and pathology. AI has been proven to be effective in reducing misdiagnoses and improving diagnostic accuracy in renal diseases. ML is expected by numerous researchers to bring drastic changes in the field of individualised diagnosis and patient treatment and is currently used to predict the nuclear grade, classification, and prognosis of RCC using radiomic data [24]. AI has also enabled the analysis of tumour sub-regions in a variety of clinical tasks, using several imaging modalities such as CT and MRI [25]. However, these analyses have been limited to only a few types of tumours, particularly brain tumours [26], head and neck tumours [27], and breast cancers [28]. To date, no study has attempted to analyse the effect of sub-region intra-tumoural heterogeneity on the diagnosis, treatment, and prognosis of renal masses and specifically ccRCCs. This rationale formed the basis of the present study, focusing on the effect of intra-tumoural heterogeneity on the grading of ccRCC. To the best of our knowledge, this is the first paper to comprehensively focus on tumour sub-regions in renal tumours for the prediction of tumour grades.

In this research, the hypothesis that radiomics combined with ML can significantly differentiate between high- and low-grade ccRCC for individual patients is tested. This study sought solutions to two major problems that previous research has not been able to address:-Characterising the effect of subregional intra-tumoural heterogeneity on grading in ccRCC;-Comparing the diagnostic accuracy of radiomics and ML through image-guided biopsy in determining the grade of renal masses, using resection (partial or complete) histopathology as a reference standard.

### Key Contributions

-Clinical Application: This research offers a practical application of radiomics and machine learning techniques in the field of oncology, specifically in the diagnosis and grading ccRCC which could potentially aid clinicians in early detection, accurate and precise treatment planning due to its higher diagnostic accuracy in comparison to traditional methods.-Subregion Heterogeneity Analysis: The inclusion of intra-tumoural subregion heterogeneity analysis highlights the depth of the research beyond simple tumour delineation and delves into the spatial distribution and variations within the tumour. This provides deeper insights into tumour biology and behaviour, leading to more personalised treatment approaches.-Potential for Non-invasive Assessment: Radiomics–machine learning algorithms, has the potential to extract valuable information from medical images non-invasively reducing the need for invasive procedures for tumour characterisation and grading, improving patient comfort and reducing healthcare costs.

## 2. Materials and Methods

### 2.1. Ethical Approval

This study was approved by the institutional board, and access to patient data was granted under Caldicott Approval Number IGTCAL11334, dated 21 October 2022. Informed consent for the research was not required, as CT scan image acquisition is a routine examination procedure for patients suspected of having ccRCC.

### 2.2. Study Cohorts

This retrospective multi-centre study used data from three centres either in partnership with or satellite hospitals of the National Health Service (NHS) in a well-defined geographical area of Scotland, United Kingdom. The institutions included Ninewells Hospital Dundee, Stracathro General Hospital, and Perth Royal Infirmary Hospital. Data from the University of Minnesota Hospital and Clinic (UMHC) were also used [29,30]. Scan data were anonymised.

We accessed the Tayside Urological Cancers (TUCAN) database for pathologically confirmed cases of ccRCC between January 2004 and December 2022. A total of 396 patients with CT scan images were retrieved from the Picture Archiving and Communication System (PACS) in DICOM format. This data formed our first cohort (cohort 1). Retrospective-based analysis for pathologically confirmed ccRCC image data following partial or radical nephrectomy from UMHC stored in a public database [31] (accessed on 21 May 2022) was performed, referred to as cohort 2 in this study. The database was queried for data between 2010 and 2018. Data for a total of 204 patients with ccRCC CT scan images was collected.

#### 2.2.1. Inclusion Criteria

-Availability of protocol-based pre-operative contrast-enhanced CT scan in the arterial phase. The selection of the arterial phase is justified due to its widespread adoption across medical centres. Moreover, this phase, characterised by its enhancement pattern and hypervascularity, has demonstrated promising capabilities in distinguishing between low- and high-grade ccRCC [32].-Confirmed histopathology from partial or radical nephrectomy with grades reported by a uro-pathologist according to the WHO/ISUP grading system.-CT scans with data to achieve a working acquisition for 3D image reconstruction.

#### 2.2.2. Exclusion Criteria

-Patients with only biopsy histopathology.-Metastatic clear cell renal cell carcinoma (mccRCC).-Patients with bilateral or ipsilateral multiple tumours, primarily due to the ambiguity of the database in distinguishing between the exact tumour grades.

For more information on the UHMC data set, the reader is referred to [29,30].

### 2.3. CT Acquisition Technique

The patients in cohort 1 were examined using up to five different CT helical/spiral scanners, including GE Medical Systems, Philips, Siemens, Canon Medical Systems, and Toshiba 512-row detectors. The detectors were also of different models, including Aquilion, Biograph128, Aquilion Lightning, Revolution EVO, Discovery MI, Ingenuity CT, LightSpeed VCT, Brilliance 64, Aquilion PRIME, Aquilion Prime SP, and Brilliance 16P. The slice thicknesses were 1.50, 0.63, 2.00, 1.25, and 1.00 mm. Likewise, the number of pixels in the image was 512 × 512. The arterial phase of the CT scan obtained 20–45 s after contrast injection was acquired using the following method: intravenous Omnipaque 300 contrast agent (80–100 mL/s), 3 mL/s contrast injection for the renal scan, and 100–120 kVp with an X-ray tube current of 100–560 mA depending on the size of the patient. For the UMHC data set, refer to [29,30].

### 2.4. Hardware and Software Consideration

A Windows 10 machine was used with the following hardware: Device name: ASUS FX503VM, Processor: Intel(R) Core(TM) i7–7700HQ CPU @ 2.80 GHz, Nvidia GeForce GTX 1060 3 GB GPU, CUDA Cores: 1152, Base Clock: 1506 MHz, Boost Clock: 1708 MHz, Texture Units: 72, Memory Clock: 8 GHz, Memory Bandwidth: 192 GB/s, ROPs: 48.L2, Cache Size: 1536 KB, Installed RAM: 32.0 GB (2 × 16 GB) DD–2400 MHz, and System type: 64-bit operating system, x64-based processor.

### 2.5. Data Curation

The procedure used for data collection with respect to each patient comprised multiple stages: accessing the Tayside Urological Cancers database, identification of patients using a unique identifier (community health index number (CHI) number), review of the medical records of the cohort, CT data acquisition, annotation of the data and, finally, quality assurance. Anonymised data for cohort 1 were in DICOM format. For each patient, duplicated DICOM slices were removed, as slice inconsistencies has detrimental effects on how an image can be processed.

Image quality is an important factor in machine learning modelling [33]. Therefore, as usual practice [32,34,35], qualitative measures were used to remove images of low quality. An expert diagnostic imaging technologist (A.J.A.) visually inspected each of the images and they were further verified by the co-authors. Images with severe blurring, granularity (quantum mottle), and ring and metal artefacts [36] were removed. Figure 1 presents a flowchart showing the exclusion and inclusion criteria of patients and their sample size distribution. Tumour grades 1 and 2 were labelled low-grade, whereas grades 3 and 4 were classified as high-grade. This is due to the clinical management for grades 1 and 2 being more or less similar, which is also the case for grades 3 and 4. A tumour classified as low-grade is more likely to grow more slowly and spread less frequently than one with a high grade.

### 2.6. Tumour Sub-Volume Segmentation Technique

In cohort 1, CT image slices for each patient were converted to 3D NIFTI (Neuroimaging Informatics Technology Initiative) format using the Python programming language version 3.9, then loaded into the 3D Slicer version 4.11.20210226 software for segmentation. Manual segmentation was performed on the 3D images, delineating the edges of the tumour slice-by-slice to obtain the VOI.

The above procedure was performed by a blinded investigator (A.J.A.) with 14 years of experience in interpreting medical images, who was unaware of the final pathological grade of the tumour. Confirmatory segmentation was carried out by another blinded investigator (A.J.) with 12 years of experience in using medical imaging technology on 20% of the samples, in order to ascertain the accuracy of the first segmentation. Thereafter, the segmentations were assessed and ascertained by an independent experienced urological surgical oncologist (G.N.), taking into consideration radiology and histology reports. The gold standard pathology diagnosis was assumed to be partial or radical nephrectomy histopathology.

For cohort 2, Heller et al. [29] previously conducted segmentation by following a set of instructions, including ensuring that the images of the patients contain the entire kidney, drawing a contour which includes the entire tumour capsule and any tumour or cyst but excluding all tissues other than renal parenchyma, and drawing a contour that includes the tumour components but excludes all kidney tissues. In the present study, only the delineation of kidney tumours achieved by Heller et al. [29] was considered. To perform delineation, a web-based interface was created on the HTML5 Canvas, which allowed contours to be drawn freehand on the images. The image series were sub-sampled in the longitudinal direction regularly, such that the number of annotated slices depicting a kidney was about 50% that of the original. Interpolation was also performed. More information on the segmentation of the cohort 2 data set can be found in the report [29].

The segmentation result for both cohorts 1 and 2 was a binary mask of the tumour. In the present study, the tumours were divided into different sub-regions based on their geometry (i.e., periphery and core). The periphery refers to regions towards the edges of the tumour, whereas the core represents regions close to the centre of the tumour. The core was obtained through extracting 25%, 50%, and 75% of the binary mask from the centre of the tumour, while the periphery was generated through extracting 25%, 50%, and 75% of the binary mask starting from the edges of the tumour to form a rim as a hollow sphere. A detailed visual description is shown in Figure 2, Figure 3 and Figure 4. Mask generation was performed using a Python script which automatically generated the sub-regions through image subtraction techniques.

### 2.7. Radiomics Feature Computation

Similar to our previous research [22], texture descriptors of the features were computed using the PyRadiomics module in Python version 3.6.1. The aim of the PyRadiomics module is to implement a standardised method for extracting radiomic features from medical images, thus avoiding inter-observer variability [37]. The parameters used in PyRadiomics were a minimum region of interest (ROI) dimension of 2, a pad distance of 5, a normalisation value of false, and a normaliser scale of 1. There was no removal of outliers, no re-sampled pixel spacing, and no pre-cropping of the image. SitkBSpline was used as the interpolator, with the bin width set to 20.

On average, PyRadiomics generated approximately 1500 features for each image and enabled the extraction of 7 feature classes per 3D image. This was performed on the 3D image in NIFTI format and the binary mask image. The extracted feature categories were as follows: first-order (19 features), grey-level co-occurrence matrix (GLCM) (24 features), grey-level run-length matrix (GLRLM) (16 features), grey-level size-zone matrix (GLSZM) (16 features), grey-level dependence matrix (GLDM) (14 features), neighbouring grey-tone difference matrix (NGTDM) (5 features), and 3D shape features (16 features). These features allow for computation of the intensity of textures as well as their distribution in the image [37].

In a previous study [22], it was shown that a combination of the original feature classes and filter features significantly improved the model performance. Therefore, we extracted the filter class features in addition to the original features. These filter classes included the local binary pattern (LBP-3D), gradient, exponential, logarithm, square-root, square, Laplacian of Gaussian (LoG), and wavelet. The filter features were applied to every feature in the original feature classes; for instance, as the first-order statistic feature class had 19 features, it follows that it had 19 LBP filter features. The filter class features were named according to the name of the original feature and the name of the filter class [37].

### 2.8. Feature Processing and Feature Selection

The features extracted using PyRadiomics were standardised to assume a standard distribution. Scaling was performed using the Z-score, as shown in Equation (Equation 1), for both the training and testing data sets independently, using only the mean and standard deviation generated from the training set. This was carried out to avoid leakage of information while also eliminating bias from the model. All of the features were transformed in such a way that they followed a standard normal distribution with mean (μ) = 0 and standard deviation (σ) = 1.
(1)Z=(x−μ)/σ,
where
*Z*: Value after scaling the feature.*x* : The feature.μ: Mean of all features in the training set.σ: Standard deviation of the training set.

Normalisation reduces the effect of different scanners, as well as any influence that intra-scanner differences may introduce in textural features, resulting in improved correlation to histopathological grade [38]. The ground-truth labels were denoted as 1 for high-grade and 0 for low-grade, for the purpose of enabling the ML models to understand the data. Machine learning models usually encounter the “curse of dimensionality” in the training data set [39], when the number of features in the data set is greater than the number of samples. Therefore, we applied two feature selection techniques in an attempt to reduce the number of features and retain only those features with the highest importance in predicting the tumour grade. First, the inter-feature correlation coefficients were computed and, when two features had a correlation coefficient greater than 0.8, one of the features was dropped. Thereafter, we used the XGBoost algorithm to further select the features with the highest importance for model development.

### 2.9. Sub-Sampling

In ML, the distribution of data among different classes is an important consideration before developing an ML model. An imbalance in the data may cause the model to become biased towards the majority class; instead of learning the features of the data, “cramming” occurs, making the model inapplicable to real-life scenarios. In this research, our data samples were imbalanced; therefore, we applied the synthetic minority oversampling technique (SMOTE) to balance the data. Care should be taken when using SMOTE, as it should only be applied in the training set and not the validation or testing sets; if this occurs, then there is a possibility that the model gains an unrealistic improvement in operational performance due to data leakage [40].

### 2.10. Statistical Analysis

Common clinical features in this research were analysed using the SciPy package. Comparisons were made based on age, gender, tumour size, and tumour volume against the pathological grade. The chi-squared test ($\chi^{2}$) was conducted to compare the associations between categorical groups. It is a non-parametric test that is used when the data do not follow the assumptions of parametric tests, such as the assumption of normality in the distribution of the data. The Student’s *t*-test is a popular statistical tool, which is used when assessing the difference between two population means for continuous data; it is normally used when the population follows a normal distribution and the population variance is unknown. The point-biserial correlation coefficient (rpb) was calculated to further confirm the significance in cases where statistical significance between clinical data was obtained. The Pearson correlation coefficient (r) was used to measure the linear correlations in the data between the radiomic features. The value of this coefficient ranged between −1 and +1, with +1 signifying a strong positive correlation.

McNemar’s statistical test, which is a modified chi-squared test, was used to test whether the difference between false negative (FN) and false positive (FP) was statistically significant. It was calculated from the confusion matrix using the stats module in the SciPy library. The chi-squared test for randomness was used to test whether the model predictions differed from random predictions. The Dice similarity coefficient was used to determine the inter-reader agreement for the segmentations. All statistical tests assumed a significance level of *p* < 0.05 (i.e., the null hypothesis was rejected when the *p*-value was less than 0.05). The radiomic quality score (RQS) was also calculated in order to evaluate whether the research followed the scientific guidelines of radiomic studies. This study followed the established guidelines of transparent reporting of a multi-variable prediction model for individual prognosis or diagnosis (TRIPOD) [41].

### 2.11. Model Construction, Validation, and Evaluation

Several models were implemented to predict the pathological grade of ccRCC, using the WHO/ISUP grading system as the gold standard. The choice of the implemented models was motivated by previous research [22,23,32,35,42,43,44,45,46,47,48,49] where the models provided satisfactory results for tumour subtype prediction using radiomics. The models were constructed for cohort 1, 2, and the combined cohort. The ML models included support vector machine (SVM), random forest (RF), extreme gradient boosting (XGBoost/XGB), naïve Bayes (NB), multi-layer perceptron classifier (MLP), long short-term memory (LSTM), logistic regression (LR), quadratic discriminant analysis (QDA), light gradient boosting machine (LightGBM/LGB), category boosting (CatBoost/CB), and adaptive boosting (AdaBoost/ADB). Different parameters were tested for each model to arrive at the optimum parameters giving the best results. Refer to Table 1 for the parameter optimisation of the models.

In total, 231 distinct models were constructed: 11 models for each of the 3 cohorts and each tumour sub-region (11 × 3 × 7). For validation, the data set was divided into training and testing sets. For the three main cohorts 1, 2 and 3; 67% of the data were used for training and 33% were retained for testing. This formed part of our internal validation procedure. Moreover, cohort 1 was validated against cohort 2 and vice versa, forming part of our external validation procedure. It should be noted that, although cohort 1 was taken to be analogous to a “single institution” data set, it was obtained from multiple centres, and its comparison with cohort 2 was for the purpose of external validation of the predictive models.

A subset of cohort 1 consisting of patients who underwent both a CT-guided percutaneous biopsy and nephrectomy (28 samples) was evaluated using two separate ML algorithms. One model was trained on cohort 1 but excluding the 28 patients who had both procedures conducted, while the second model was trained on cohort 2, acting as an external validator. The classifiers and sub-regions were determined for the two best-performing classifiers and the three best-performing tumour regions. The objective of this test was to assess the accuracy of tumour grade classification in biopsy histopathology when compared to ML prediction using partial or total nephrectomy histopathology as the gold standard. These 28 samples are referred to as cohort 4. In situations where the biopsy grade histopathology was indeterminate for a specific tumour, we concluded its final pathological grade as the opposite of the nephrectomy grade for that tumour (i.e., if the nephrectomy outcome was high-grade but the biopsy result was indeterminate, we concluded that the biopsy was low-grade for the purpose of analysis). It is worth-noting that “Indeterminate results”are important as they do not contribute to decision-making and one of the reasons biopsy approaches has not been adopted amongst clinician’s world over. In fact, our group addressed this issue by 3 × 2 tables in a systematic review published earlier [50]. We believe that all studies should report “indeterminate results” for the sake of transparency and external validity.

Evaluation of the model performance was carried out using a number of metrics, including accuracy (ACC), specificity (SPE), sensitivity (SEN), area under the curve of the receiver operating characteristic curve (AUC-ROC), the Matthews correlation coefficient (MCC), F1 score, McNemar’s test (McN), and the chi-squared test ($\chi^{2}$). All major metrics were reported at 95% Confidence Interval (CI).

In a highly imbalanced data set, accuracy is not reliable as it gives an overly optimistic measure of the majority class [51]. MCC is an effective solution to overcome class imbalance and has been applied by the Food and Drug Administration (FDA) agency in the USA for the evaluation of MicroArrayII/Sequencing Quality control (MAQC/SEQC) projects [52]. There are situations, however, where MCC gives undefined or large fluctuations in the results [53]. The combination of precision/recall, which is the F1-score [54], provides better information than the pair of sensitivity/specificity [55] and has gained popularity since the 1990s in the machine learning world. Despite its popularity, the F1-score varies for class swapping while the MCC is invariant. The AUC-ROC curve is a popular evaluation metric used when a single threshold of the confusion matrix is unavailable [56]. It is also sensitive to class imbalance, though it is widely used in the medical field; therefore, it was used in this study to compare with previous research. All the other metrics have been reported as well and it is at the readers discretion to compare which metrics suit the study.

## 3. Results

### 3.1. Study Population and Statistical Analysis

In cohort 1, after implementation of the inclusion and exclusion criteria, a total of 187 patients with pathologically proven ccRCC were obtained. Of these, 80 patients presented low-grade and 107 presented high-grade ccRCC. The mean age was 59.05 and 64 years for low- and high-grade tumours, respectively. Gender-wise, 65.78% of patients were male and 34.22% were female. The average tumour size and tumour volume were 4.32 cm and 75.8 cm^3^, respectively, for low-grade patients, and 6.033 cm and 203.74 cm^3^, respectively, for high-grade patients. For cohort 2, the data set met all the inclusion and none of the exclusion criteria; hence, no sample was eliminated. The mean age was 57.17 and 63.68 years for low- and high-grade patients. The average tumour size and tumour volume were 3.89 cm and 51.44 cm^3^ for low-grade patients, and 6.81 cm and 235.26 cm^3^ for high-grade patients, respectively. In terms of gender, 65.69% of patients were male, while the rest were female.

Differences in average age, tumour size, and tumour volume (but not gender) were statistically significant in cohorts 1, 2, and the combined cohort. However, using the point-biserial correlation coefficient (rpb) to compare the correlation between the best model prediction and the clinical features, no statistically significant difference was found. Table 2 provides the characteristics and analysis of patients. The Dice similarity coefficient score was 0.93, which indicated that there was a good inter-reader agreement for tumour segmentation. The entire data set RQS was found to be 61.11%, signifying that the research followed scientific radiomic guidelines. For the RQS rubric, we refer the reader to https://www.radiomics.world/rqs2 (accessed on 27 April 2023) [41].

### 3.2. Feature Extraction, Pre-Processing, and Selection

A total of 1875 features were extracted using the PyRadiomics library. It should be noted that there were no null values in the data, as it is crucial in the context of ML to handle these values to avoid errors and undefined results. The Pearson correlation coefficient (r) and extreme gradient boosting algorithm were used to reduce the number of features to an optimal number. The number of features selected (FS) varied between the data sets.

### 3.3. Model Validation and Evaluation

#### 3.3.1. Internal Validation

**Cohort 1**: Of the developed models, the CatBoost classifier performed the best for the majority of the tumour sub-region models, with its best classifier having an AUC of 85.0% in the 100% tumour. When the tumour sub-region was considered, the 50% tumour core and 25% tumour periphery exhibited the best performance, with an average AUC of 77.9% and 78.6%, respectively. When the models’ core and periphery regions were averaged, the best classifier was CatBoost, with an AUC of 80.0% = (80.7% + 79.3%)/2. Table A1 and Table A2 provide the results obtained for cohort 1.

**Cohort 2**: The best-performing model in the cohort 2 data set was the CatBoost classifier, with the best performance in the 50% tumour periphery having an AUC of 91.0%. In terms of tumour sub-region, the 50% tumour core had the highest average AUC of 87.6%. When the models’ subregions were averaged, the best classifier was CatBoost, with an AUC of 86.5% = (87.0% + 86.0%)/2. Table A3 and Table A4 provide the results obtained for cohort 2.

**Cohort 3**: When NHS and UMHC data were combined, the models with the highest AUC were the 50% tumour core CatBoost classifier and the 75% tumour periphery RF classifier, with AUC of 80.0% for both. The 50% tumour core was the best region, with an average AUC of 76.9%. When the models’ core and periphery regions were averaged, the best classifiers were RF and CatBoost, with AUC values of 77.3% = (76.3% + 78.3%)/2 and 77.0% = (77.3% + 76.7%)/2, respectively. Table A5 and Table A6 provide the results obtained for cohort 3.

#### 3.3.2. External Validation

**Cohort 1**: When cohort 2 was used as the training set and cohort 1 was predicted on its models, the best-performing model was the QDA 25% tumour periphery classifier, with an AUC of 71.0%. For the tumour sub-region, the 25% tumour periphery was the best, with an average AUC of 65.0%. When the models were averaged, the best classifier was QDA, with an AUC of 67.7% = (67.3% + 68.0%)/2. Table A7 and Table A8 provide the relevant results.

**Cohort 2**: With cohort 1 as the training set and cohort 2 as the testing set, the best-performing model was the SVM 50% tumour core classifier, with an AUC of 77.0%. In terms of tumour sub-region, the 50% tumour core was the best, with an average AUC of 74.2%. When the models were averaged, the best classifier was RF, with an AUC of 74.8% = (74.3% + 75.3%)/2. Refer to Table A9 and Table A10 for the results.

### 3.4. Comparison between Biopsy and Machine Learning Classification

When the biopsy classification results on the 28 samples separated from the NHS data set were compared to ML predictions, machine learning exhibited the highest AUC values of 95.0% and 80.0% for internal and external validation, respectively, using the CatBoost classifier. This was better than the AUC of 31.0% obtained from biopsy results, in terms of correctly grading renal cancer, as shown in Figure A1. Relevant statistics are provided in Table 3 and Table A11.

## 4. Discussion

Clear cell renal cell carcinoma is the most common subtype of renal cell carcinoma and is responsible for the majority of renal cancer-related deaths. It comprises up to 80% of RCC diagnoses [5], and is more likely to metastasise to other organs. Important diagnostic criteria that must be derived include tumour grade, tumour stage, and the histological type of the tumour. For most cancer patients, histological grade is a crucial predictor of local invasion or systemic metastases, which may affect how well they respond to treatment. To define the extent of the tumour, tumour staging-based clinical assessment, imaging investigations, and histological assessment are required. A greater comprehension of the neoplastic situation and awareness of the limitations of diagnostic techniques are made possible through an understanding of the procedures involved in tumour diagnosis, grading, and staging.

To accurately grade a tumour, several grading schemas have been applied, of which the WHO/ISUP and Fuhrman grading systems are the most popular and widely accepted. Previously, grading was focused on a collection of cytological characteristics of the tumour; however, nuclear morphology has more recently become a major area of focus. The Fuhrman grading system has been used for some time [57], with its worldwide adoption in 1982 [2].

There are several methodological issues with the study conducted by Fuhrman et al.; for example, its reliance on retrospective data collected over a 13-year period raises questions about potential biases [1]. The system’s dependency on a small sample size of only 85 cases may also make its conclusions less generalisable [1,57]. The inclusion of several RCC subtypes without subtype-specific grading eliminated the possibility of variations in tumour behaviour [1,57,58]. It is difficult to grade consistently and accurately, due to the complexity of the criteria, which call for the simultaneous evaluation of three nuclear factors (i.e., nuclear size, nuclear irregularity, and nucleolar prominence) [57,58], resulting in poor inter-observer reproducibility and interpretability. The lack of guidelines that can be utilised to assign weights to the different discordant parameters to achieve a final grade makes the Fuhrman system even more controversial [57,59]. Furthermore, the shape of the nucleus has not been well-defined for different grades [57]. Grading discrepancies are a result of conflict between the grading criteria and a lack of direction for resolving them [1,3,58]. Additionally, imprecise standards for nuclear pleomorphism and nucleolar prominence adversely affect classifications made by pathologists, resulting in increased variability [57]. Even if a tumour is localised, grading according to the highest-grade area could result in an over-estimation of tumour aggressiveness [1,57]. This system’s inconsistent behaviour and poor reproducibility [58] have raised questions regarding its dependability and potential effects on patient care and prognosis [60]. Flaws regarding inter-observer repeatability [60], and the fact that the Fuhrman grading system is still widely used despite these flaws, indicate that there is a need for more research and better grading methods.

An extensive and co-operative effort resulted in the development of the ISUP grading system for renal cell neoplasia in 2012 as an alternative to the Fuhrman grading system [57,58]. The system was ratified and adopted by the WHO in 2015 and renamed as the WHO/ISUP grading system [1,4]. As opposed to the Fuhrman grading system, the ISUP system focuses on the prominence of nuclei as the sole parameter that should be utilised when identifying the tumour grade. This reduction in rating parameters has led to better grade distinction and increased predictive value. This has also eliminated the controversy around reproducibility that had been identified with respect to the Fuhrman grading system. Previous studies have shown that there is a clear separation between grades 2 and 3 in the WHO/ISUP grading system, which was not the case with the Fuhrman system. Indeed, in their study, Dugher et al. [3] have highlighted the downgrade of Fuhrman grades 2 and 3 to grades 1 and 2, respectively, in the WHO/ISUP system. This indicates that, besides the overlap of grades in the Fuhrman system, there was also an over-estimation of grades—a problem that has been rectified with the WHO/ISUP grading system [3,23]. The WHO/ISUP grading system has been highly associated with the prognoses of patients.

Pre-operative image-guided biopsy is a diagnostic tool that is used to identify the tumour grade. However, there are inherent problems that have been identified in connection with this approach, including the fact that it is invasive in nature, causes discomfort, and may lead to other complications in patients when the procedure is performed [12]. Therefore, non-invasive testing, imaging, and clinical evaluations may be necessary to confirm the presence of ccRCC and its grade without having to undergo such a procedure. Radiomics has gained traction in clinical practice in recent years, and has been a buzzword since 2016 [20]. It refers to the extraction of high-dimensional quantitative image features in what is known as image texture analysis, describing the pixel intensity in medical images such as X-ray images as well as CT, MRI, CT/PET, CT/SPECT, US, and mammogram scans. Radiomics approaches have been applied in a number of studies for the diagnosis, grading, and staging of tumours. Machine learning is one of the major branches of AI, providing methods that are trained on a set of known data and then tested on unknown data. In this way, researchers have attempted to make machines more intelligent through determining spatial differences in data that would have been otherwise difficult for a human being to decipher. Such approaches have been used in combination with texture analyses, particularly for tumour classification, grading, and staging. They are capable of learning and improving through the analysis of image textural features, thereby resulting in higher accuracy than native methods [61].

Heterogeneity within tumours is a significant predictor of outcomes, with greater diversity within the tumour being potentially linked to increased tumour severity. The level of tumour heterogeneity can be represented through images known as kinetic maps, which are simply signal intensity time curves [62,63,64]. Previous studies [65,66] that have utilised these maps typically end up averaging the signal intensity features throughout the solid mass; hence, regions with different levels of aggressiveness end up contributing equally to determining the final features. This leads to a loss of information regarding the correct representation of the tumour [67,68]. In some studies, there have been attempts to preserve intra-tumoural heterogeneity through extracting the features at the periphery and the core and analysing them separately [17,18,19,66]; however, this is still not sufficient, as information from other sub-regions of the tumour is not considered.

The objective of this work was to study the impact of subregion intra-tumoural heterogeneity on the grading of ccRCC, comparing and contrasting ML/AI based methods combined with CT radiomics signatures with biopsy and partial and radical nephrectomy in terms of determining the grade of ccRCC. Finally, the possibility of using CT radiomics ML analysis as an alternative to—and, thereby, as a replacement for—the conventional WHO/ISUP grading system in the grading of ccRCC was investigated.

The experimental findings of our research highlighted various aspects for discussion. From the results, it was found that age, tumour size, and tumour volume were statistically significant for cohorts 1, 2, and 3. However, for cohort 4, none of the clinical features were found to be significant. Upon further analysis of the statistically significant clinical features using the point-biserial correlation coefficient (rpb), no features were verified as significant. Furthermore, the 50% tumour core was identified as the optimal tumour sub-region, exhibiting the highest average performance across models in cohorts 1, 2, and 3, with average AUCs of 77.9%, 87.6%, and 76.9%, respectively. It is worth noting that the 25% tumour periphery presented an increase in average performance for cohort 1, having an AUC of 78.6%; however, this result was not statistically different from that of the 50% core, and it failed to register the best performance in the other cohorts.

Among the 11 classifiers, the CatBoost classifier was the best model in all three cohorts, with average AUC values of 80.0%, 86.5%, and 77.0% for cohorts 1, 2, and 3, respectively. Likewise, the best-performing distinct classifier per cohort was CatBoost, with AUC of 85.0% in the 100% core, 91.0% in the 50% periphery, and 80.0% in the 50% core for cohorts 1, 2, and 3, respectively. In the external validation, cohort 1 validated on cohort 2 data had the highest performance in the 25% periphery, with the highest AUC of 71.0% and the best classifier being QDA. Conversely, cohort 2 validated on cohort 1 data provided the best performance in the 50% core, with an AUC of 77.0% and the best classifier being the SVM. Finally, in the comparison between biopsy- and ML-based classification of the 28 patients who underwent both biopsy and nephrectomy (i.e., cohort 4), the ML model was found to be more accurate, with the best AUC values for internal and external validation being 95.0% and 80.0%, respectively, in comparison to an AUC of 31.0% when biopsy was performed. In this case, the nephrectomy results of grading were assumed as the ground-truth.

For each of the 231 models the pathological grade of a tumour was predicted in less than 2 s. It is worth noting that the segmentations in cohort 1 were markedly different from those in cohort 2. Cohort 2 emerged as the highest-performing group, followed by cohort 1, while the combined cohort, notably cohort 3, exhibited the lowest performance. This disparity can be attributed to several factors, including variations in scanners, segmentations, pixel size, section thickness, tube current, tube voltage, kernel reconstruction, enhancement of contrast agent and imaging protocols. Moreover, cohort 1, in itself is a multi-institutional data set from three different centres. This may contrebuted to the low performance during external validation. Refer to Table A1, Table A2, Table A3, Table A4, Table A5, Table A6, Table A7, Table A8, Table A9 and Table A10 for comparison.

Clinical feature significance is an important aspect of any research, as it gives a general overview of the data to be used in a study. Few studies have opted to include clinical features which are statistically significant into their ML radiomics models [69,70]. Takahashi et al. [70], for instance, incorporated 9 out of 12 clinical features into their prediction model due to them being statistically significant [70]. In our study, age, tumour size, and tumour volume were found to be statistically significant; however, they were not integrated into the ML radiomics model as a confirmatory test using the point-biserial correlation coefficient revealed a lack of significance. Nonetheless, there is a lack of clear guidelines on the relationship between statistical significance and predictive significance. There is a misunderstanding that association statistics may result in predictive utility; however, association only provides information regarding a population, whereas predictive research focuses on either multi-class or binary classification of a singular subject [71]. Moreover, the degree of association between clinical features and the outcome is affected by sample size; that is, statistical significance is likely to increase with an increase in sample size [72]. This has been clearly portrayed in previous research, such as that of Alhussaini et al. [22]. Even in our own research, for the cohort 4 data—despite being derived from the same population as cohort 1—the age, tumour size, tumour volume, and gender were not statistically significant, indicating that the sample size might be the likely cause.

### 4.1. Literature Related to Methodological Proposal

Zhao et al. [43], in their prospective research, presented interesting findings regarding tumour sub-regions in ccRCC. In their research, they indicated that somatic copy number alterations (CANs), grade, and necrosis are higher in the tumour core, compared to the tumour margin. Our findings, obtained using different tumour sub-regions, tend to agree with the study by Zhao et al. [43], even though the authors did not construct a predictive ML algorithm.

He et al. [44] constructed five predictive CT scan models using an artificial neural network algorithm, in order to predict the tumour grade of ccRCC using both conventional image features and texture features. The best-performing model in their study, using the corticomedullary phase (CMP) and the texture features, provided an accuracy of 91.8%. This is comparable to our study, in which the CatBoost classifier attained the highest accuracy of 91.1%. However, He et al. [44] did not use other metrics, which could have been useful in analysing the overall success of the prediction. For instance, the research could have depicted a high accuracy but with bias towards one class. Moreover, the research findings were not externally validated; hence, its performance is unclear with respect to other data sets.

Similar to He et al. [44], Sun et al. [35] constructed an SVM algorithm to predict the pathological grade of ccRCC. The results of their research gave an AUC of 87.0%, sensitivity of 83.0%, and specificity of 67.0%. However, we found that they erred by giving an overly optimistic AUC with a very low specificity. This can easily be seen by analysing our SVM results for the best-performing SVM model, which had an AUC of 86.0%, sensitivity of 81.0%, and specificity of 91.5%. Our best model—the CatBoost classifier—performed much better.

Xv et al. [45] set out to analyse the performance of the SVM classifier using three feature selection algorithms for the differentiation of ccRCC pathological grades in both clinical–radiological and radiomics features. The three algorithms were the LASSO, recursive feature elimination (RFE), and ReliefF algorithms. Their best-performing model was SVM–ReliefF with combined clinical and radiomics features, with an AUC of 88.0% in the training set, 85.0% in the validation set, and 82.0% in the test set. It is worth noting that we used none of the feature selection algorithms used by Xv et al. [45], but obtained better performance.

Cui et al. [46] conducted internal and external validation for the purpose of predicting the pathological grade of ccRCC. Their research achieved satisfactory performance, with internal and external validation accuracy of 78.0% and 61.0%, respectively, in the corticomedullary phase using the CatBoost classifier. Compared to their research, our findings indicated better performance when the CatBoost classifier was used for both the internal and external validation, with an accuracy of 91.2% and 76.0%, respectively, in the CMP.

Wang et al. [47] also conducted a multi-centre study using a logistic regression model; however, they used both biopsy and nephrectomy as the ground-truth, despite the challenges that have been highlighted regarding biopsies. They did not report on the internal validation performance; however, their training AUC, sensitivity, and specificity were 89.0%, 85.0%, and 84.0%, respectively. Likewise, their external validation AUC, sensitivity, and specificity were 81.0%, 58.0%, and 95.0%, respectively. Their external validation performance was better than our performance using the LR model, which obtained an AUC, sensitivity, and specificity of 74.0%, 59.7%, and 88.2%, respectively. However, in general, our CatBoost classifier still out-performed their LR model.

Moldovanu et al. [48] investigated the use of multi-phase CT using LR to predict the WHO/ISUP nuclear grade of ccRCC. When our results were compared with their validation set, which yielded an AUC, sensitivity, and specificity of 81.0%, 72.7%, and 75.9% in the corticomedullary phase, our research exhibited higher performance not only in the best-performing model but also in the LR model, which obtained an AUC, sensitivity, and specificity of 84.0%, 71.4%, and 95.8%, respectively.

Yi et al. [49] have performed research for prediction of the WHO/ISUP pathological grade of ccRCC using both radiomics and clinical features with an SVM model. The 264 samples used were from the nephrographic phase (NP). We noted that there was a massive class imbalance in the data, with a ratio between low- and high-grade samples of 78:22; however, they did not highlight how this issue was resolved. Nonetheless, the testing accuracy yielded an AUC of 80.2%, lower than that obtained in our research.

Similar to our study, Karagöz and Guvenis [42] constructed a 3D radiomic feature-based classifier to determine the nuclear grade of ccRCC using the WHO/ISUP grading system. The best results were obtained using the LightGBM model, which obtained an AUC of 0.89. They also carried out tumour dilation and contraction by 2 mm, which led them to conclude that the ML algorithm is robust against deviation in segmentation by observers. Our best model out-performed their research and our sample size was much larger, thereby providing more trustworthy results.

Demirjian et al. [23] also constructed a 3D model using data from two institutions using RF, AdaBoost, and ElasticNet classifiers. The best-performing model, RF, obtained an AUC of 0.73. This model performance was lower than in our research. The use of a data set graded using the Fuhrman system for testing may have led to poor results, as WHO/ISUP and Fuhrman use different parameters for grading; hence, it is not advisable to use the Fuhrman grade as the ground-truth for a model trained using WHO/ISUP.

Shu et al. [32] extracted radiomics features from the CMP and NP to construct 7 ML algorithms, with the best model in the CMP (i.e., the MLP algorithm) achieving an accuracy of 0.97. The findings of this study are quite interesting, but the gold standard used for grade prediction was not discussed; this may lead us to the conclusion that biopsy was part of the gold standard. We have highlighted the controversies surrounding biopsies and, accordingly, the research may have been affected by such issues. There are some studies which have applied deep learning for the prediction of tumour grade [73,74,75]. The AUC in these studies ranged from 77.0% to 88.2%. These results are not only worse than those obtained in the current research, but the Fuhrman grading system was also used as the gold standard.

### 4.2. Biopsy Grading and Its Comparison with ML

A biopsy Biopsy is a commonly used diagnostic tool for the identification of RCC subtypes. The diagnostic accuracy of biopsy for RCC has been reported to range from 86.0 to 98.0%, but this can be influenced by various factors [12,76,77]. Notably, when it comes to grading RCC, the range of accuracy widens to between 43.0 and 76.0% [12,76,77,78,79,80,81,82,83]. Nevertheless, a biopsy’s accuracy in classifying renal cell tumours is debatable (Millet et al., 2012 [80]). Different studies have contended that a kidney biopsy typically understates the final grade. For instance, biopsies underestimated the nuclear grade in 55% of instances and only properly identified 43% of the final nuclear grades [78]. In particular, the final nuclear grade was marginally more likely to be understated in biopsies of larger tumours, while histologic subtype analysis yielded more accurate results; especially when evaluating clear cell renal tumours. In the research by Blumenfeld et al. [78], only one case of the nuclear grade being over-estimated was reported. In the study of Millet et al. [80], biopsy led to under-estimation of the grade in 13 cases while, in 2 cases, it over-estimated the grade.

In our study, we found that the accuracy of biopsy was 35.7% in determining the tumour grade, with a sensitivity and specificity of 9.1% and 52.9%, respectively, in the 28 NHS samples (cohort 4) when nephrectomy was used as a gold standard as shown in Table 3. These results are in agreement with previous studies [78,80] which determined biopsy to perform poorly in grading tumours. The results obtained through biopsies were compared to our ML models, and the models out-performed biopsies by far; in fact, our worst-performing model was still better than biopsy. The best model had an accuracy of 96.4%, sensitivity of 90.9%, and specificity of 100% in the internal validation, comprising a 60.7% improvement in accuracy. Likewise, in the external validation, there was a 46.4% improvement in accuracy, with an accuracy, sensitivity, and specificity of 82.1%, 72.7%, and 88.2%, respectively as presented in Table A11. Therefore, we can conclude that ML approaches are able to distinguish low- from high-grade ccRCC with better accuracy, when compared to biopsy, and thus should be considered as a replacement.

In previous research, no paper has tackled the effect of tumour sub-region with regard to the grading of ccRCC; hence, there were no studies with which our results could be compared. The current research dived deeper into the possibility of pre-operatively grading ccRCC without the need for biopsy. Moreover, the effect of the information contained in different tumour sub-regions on grading was analysed. It is the belief of the authors of this research that the results of this study will assist clinicians in finding the best management strategies for patients of ccRCC, as well as enabling informative pre-treatment assessments that allow treatments to be tailored to individual patients.

## 5. Limitations, Takeaways and Summary

### 5.1. Limitations and Future Research

The work encountered a few challenges which are important to highlight. The samples used in this study were obtained from different institutions, and the scans were captured using different scanners and protocols. This may have lowered the overall performance of the models. However, it was important to use such data as the research was not meant to be institution-specific but, instead, generally applicable. Second, the retrospective nature of the research may have limited our work, and it is therefore recommended that more research should be conducted through a prospective study. Third, the current research assumed that the divided tumour sub-regions (25%, 50%, and 75% core and periphery) are heterogeneous in nature. In this regard, more research using pixel intensity measurement from different tumour sub-regions is encouraged. Fourth, manual segmentation is not only time-consuming but also subject to observer variability; thus, research on automated tumour image segmentation techniques is encouraged. Fifth, the predominant approach to grading ccRCC studies revolves around utilising a binarised model output. This is motivated by two primary factors. First, there exists a notable discrepancy in the sample sizes across different grades, with grades 1 and 4 exhibiting smaller sample size compared to grades 2 and 3. Second, adopting a 4-class model is perceived to yield minimal impact on patient management, given the similarity in management strategies between low grades (I and II) and high grades (III and IV) [84]. Nonetheless, there is merit in exploring the application of a 4-class model in forthcoming investigations, as doing so may validate the suitability of radiomics machine learning analyses in delineating distinct WHO/ISUP grading categories. Moreover, despite this being one of the few studies which has used a large sample size, we still consider our sample size to be small with respect to ML and AI approaches, which often require larger data sets for training. Finally, it’s advisable to undertake a deep learning research using a substantial data set based on the WHO/ISUP grading systems.

#### 5.1.1. Take-Home Messages

-Radiomics features combined with ML algorithms have the potential to predict the WHO/ISUP grade of ccRCC more accurately than pre-operative biopsy.-Analysing different tumour subregions, such as the 50% tumour core and 25% tumour periphery, provides valuable information for determining tumour grade.-Analysing different cohorts from both single and multi-centre studies represented the effect of data heterogeneity on the model’s performance. This underscores the importance of implementing a robust model that generalises well for real-world applications in grading ccRCCs.-The study highlighted the promising application of advanced imaging techniques and ML in oncology for precise tumour grading.

#### 5.1.2. Summary

In this study, an in-depth radiomics ML analysis of ccRCC was carried out with the purpose of determining the clinical significance of intra-tumoural sub-region heterogeneity in CT scans and biopsies with respect to the accuracy of tumour grading. In this regard, the results support the assertion that tumour sub-regions are an important factor to consider while grading ccRCC. We were able to demonstrate that the 50% tumour core can be considered the best sub-region for determining the tumour grade; however, this should not be interpreted as indicating that other tumour sub-regions are unimportant. Indeed, the results indicated only small differences in performance for the different tumour sub-regions; therefore, the different regions should be analysed independently and taken into consideration for the final grading outcome. Regarding the second objective on the importance of biopsy in grading, through comparison of our research results with biopsy results, we were able to demonstrate that ML approaches yield much better results in terms of determining the ccRCC WHO/ISUP grade. Finally, the performance of the ML models in determining tumour grade demonstrates the potential benefit of using ML as an alternative or replacement for biopsy in determining the tumour grade.

## 6. Conclusions

In conclusion, the present work demonstrated the potential of ML models for distinguishing low- from high-grade ccRCC. In essence, ML approaches can act as a “virtual biopsy,” being potentially far superior to biopsy for grading purposes. These findings have important clinical significance for addressing the challenges that are experienced in relation to biopsies, leading to improved clinical management and contributing to oncological precision medicine. 

## Figures and Tables

**Figure 1 cancers-16-01454-f001:**
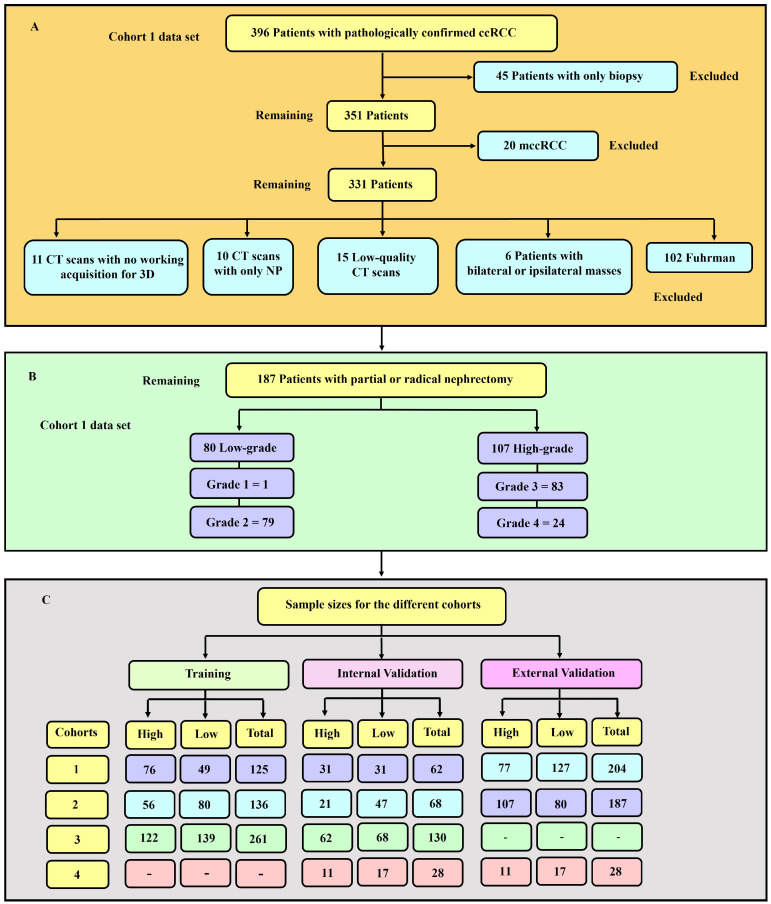
(**A**) Diagrammatic representation of the exclusion and inclusion criteria for the cohort 1 data set. 45 patients with only biopsy and 20 with mccRCC were excluded. (**B**) Distribution of sample sizes in cohort 1, categorised by low- and high-grade. There was 80 and 107 low- and high-grade tumours respectively. (**C**) Sample sizes for the various cohorts utilised in this study. The training, internal and external validation samples for each of the four cohorts are highlighted.

**Figure 2 cancers-16-01454-f002:**
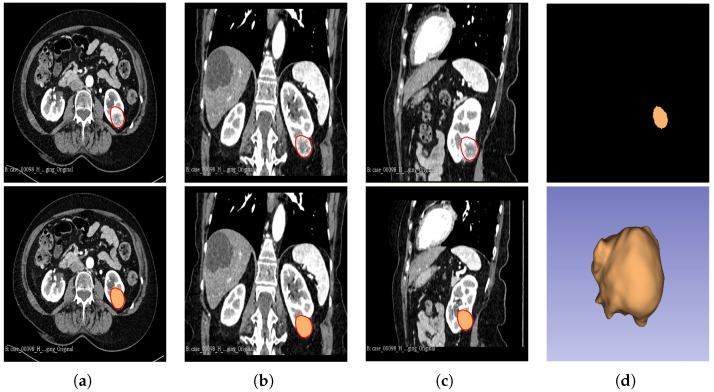
Manual segmentation of the 3D image slices of the tumour as presented by the red area was performed using the 3D Slicer software: version 4.11.20210226 (**a**) axial plane; (**b**) coronal plane; (**c**) sagittal plane; and (**d**) generated 3D VOI from the delineated 2D slices.

**Figure 3 cancers-16-01454-f003:**
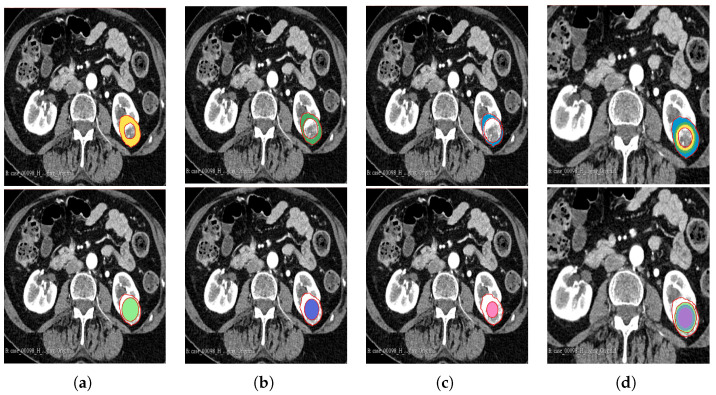
Manual segmentation of the 3D image slices using the 3D Slicer software: version 4.11.20210226 (**a**) 75% periphery (yellow) and core (green) of the tumour; (**b**) 50% periphery (green) and core (purple) of the tumour; (**c**) 25% periphery (blue) and core (pink) of the tumour; and (**d**) overlap of the periphery and core sub-regions.

**Figure 4 cancers-16-01454-f004:**
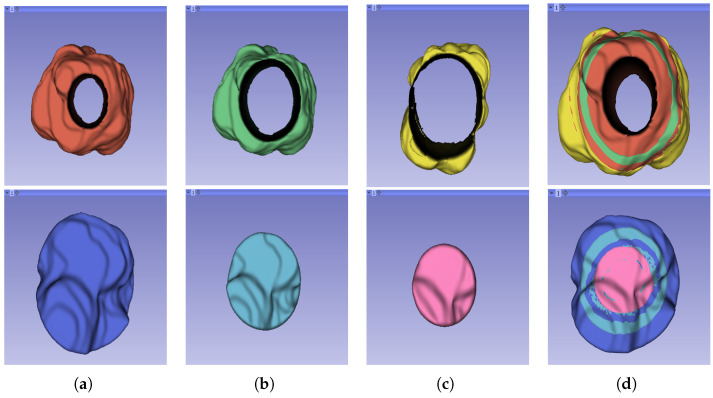
Representation of the 3D segmented regions: (**a**) 75% periphery (red) and core (blue) of the tumour; (**b**) 50% periphery (green) and core (blue) of the tumour; (**c**) 25% periphery (yellow) and core (pink) of the tumour; and (**d**) overlap of the periphery and core sub-regions.

**Table 1 cancers-16-01454-t001:** Parameter optimisation for machine learning models.

Models	Parameters
SVM	kernel = rbf, probability = True, random_state = 42, gamma = 0.2, C = 0.01.
RF	n_estimators = 401, random_state = 42, max_depth = 3.
XGBoost	random_state = 42, learning_rate = 0.01, n_estimators = 401, gamma = 0.52.
NB	GaussianNB.
MLP	hidden_layer_sizes = (401,201), activation = relu, solver = adam, max_iter = 5.
LSTM	loss = binary_crossentropy, optimizer = Adam, lr = 0.01, metrics = accuracy, epochs = 1000, batch_size = 16.
LR	random_state = 42, max_iter = 4.
QDA	reg_param = 0.05.
LightGBM	random_state = 42, n_estimators = 9.
CatBoost	random_state = 42, verbose = False, iterations = 50.
AdaBoost	base_estimator = rf, n_estimators = 201, learning_rate = 0.01, random_state = 42.

**Table 2 cancers-16-01454-t002:** Statistical demographic characteristics of patient data.

Tumour and Patient Characteristics					
	Variable	Low-Grade	High-Grade	*p*-Value	rpb *^1^
cohort 1 n = 187	Age (Mean ± SD)	59.05 ± 12.28	64 ± 9.40	0 *	0.4
	Size (cm)	4.32 ± 2.02	6.03 ± 3.23	0 *	0.6
	Volume (cm^3^)	75.8 ± 90.90	203.74 ± 305.82	0 *	0.6
	Gender			0.331	
	Male	49 (26.20%)	74 (39.57%)		
	Female	31 (16.58%)	33 (17.65%)		
cohort 2 n = 204	Age (Mean ± SD)	57.17 ± 12.67	63.68 ± 11.14	0 *	0.88
	Size (cm)	3.89 ± 2.16	6.81 ± 3.55	0 *	0.45
	Volume (cm^3^)	51.44 ± 114.32	235.26 ± 326.10	0 *	0.56
	Gender			0.25	
	Male	77 (37.75%)	57 (27.94%)		
	Female	50 (24.51%)	20 (9.80%)		
cohort 3 n = 391	Age (Mean ± SD)	57.89 ± 12.55	63.86 ± 10.17	0 *	0.56
	Size (cm)	3.89 ± 2.16	6.81 ± 3.55	0 *	0.2
	Volume (cm^3^)	60.86 ± 106.55	216.93 ± 314.85	0 *	0.29
	Gender			0.25	
	Male	126 (32.23%)	131 (33.50%)		
	Female	81 (20.72%)	53 (13.55%)		
cohort 4 n = 28	Age (Mean ± SD)	57.12 ± 10.25	62.09 ± 9.39	0.22	
	Size (cm)	3.31 ± 0.94	4.02 ± 2.25	0.28	
	Volume (cm^3^)	25.98 ± 27.10	57.16 ± 70.40	0.13	
	Gender			1	
	Male	12 (42.86%)	8 (28.57%)		
	Female	5 (17.86%)	3 (10.71%)		

* Statistical significance at the 0.05 level; *^1^ point-biserial correlation coefficient (rpb).

**Table 3 cancers-16-01454-t003:** Comparison of the best diagnostic performance in cohort 4 under biopsy and machine learning models.

	Biopsy	Machine Learning	
Metrics		Internal Validation	External Validation
ACC	35.71 ± 17.75	96.43 ± 5.22	82.14 ± 14.19
SPE	52.94 ± 23.73	100.0 ± 12.13	88.24 ± 13.54
SEN	9.09 ± 13.04	90.91 ± 13.04	72.73 ± 26.32
AUC	31.0 ± 17.1	95.0 ± 8.1	80.0 ± 14.8
MCC	−0.40	0.93	0.62
F1	0.1	0.95	0.76
McN	0.64	0.32	0.65
$\chi^{2}$	0.15	0	0.06

## Data Availability

The data provided in the cohort 1 study are available on request from the corresponding authors. For cohort 2, the data are readily available from the Kits GitHub page and Cancer Imaging Archive (CIA) [29,30,31]. The codes used to reproduce the results can be found on GitHub, upon request, at the following link: https://github.com/abeer2005/Prediction-Clear-Cell-Renal-Carcinoma-Grade (accessed on: 15 December 2023).

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
