# Peer review of "Radiomics Machine Learning Analysis of Clear Cell Renal Cell Carcinoma for Tumour Grade Prediction Based on Intra-Tumoural Sub-Region Heterogeneity"

_cancers, 2024, doi:10.3390/cancers16081454_

Round 1

Reviewer 1 Report (Previous Reviewer 2)

Comments and Suggestions for Authors

Introduction:

- Still a lot of information that does not seem to be necessary to review for the study. The tumor characteristics of RCC tumors described in paragraph 2 seem excessive. Trimming down this section would be best

- Better outline for the goals and objectives of the paper

- Grammar seems much better than the previous submission

Materials and Methods:

- Inclusion and exclusion criteria are more detailed and clearly describe why specific patients were excluded

- Description of “low quality CT scans” is qualitative, not quantitative. What specific quality control measures did you use to justify excluding these patients? Or was it all done by subjective viewers? If you did use quantitative measures to make these decisions, what were the cutoffs for SNR and contrast levels? What resolution did you use as a cutoff to remove patients? These are important details.

- The smoothing process seems unnecessary, how much did this affect your results? Did it improve them significantly or can your model stand without them? How much did your radiomics features change with this addition?

- Somewhat strange validation procedure. Would have been clearer to take Cohort 1 as a training/internal validation set, and then cohort 2 as an external validation set.

- Training/validating on both cohorts separately is strange, and training models on both sets is also strange. You don’t have a true holdout validation set in this setup, since all the data is being used in the training process at some point.

- The approach to the biopsy analysis is unclear. Did you just use a different ground truth for this dataset based on the biopsy evaluation instead of the PN/TN? Or did you actually train a model using the biopsies to predict grade? What is the motivation behind labeling indeterminate biopsy results as the OPPOSITE of the nephrectomy? This seems to obviously bias the results towards being worse for this dataset.

Results:

- It is very questionable that biopsies would have only a 31% AUC in grading renal cancer. From my knowledge, this is not consistent with prior literature in this area. I think the unclear explanation of this in the methods is also not helping

Comments on the Quality of English Language

Improved from previous versions

Author Response

N/A

Reviewer 2 Report (New Reviewer)

Comments and Suggestions for Authors

Thank you for the opporunity to review this interesting paper.

I have only small concerns regarding this work.

-way too long introduction

-provide 95% CI for your results, especially the AUC values

-table 3 needs clarifications and abbreviations explanations

-discussion also way too long

142 references are way too much for a research article

Author Response

N/A

Round 2

Reviewer 1 Report (Previous Reviewer 2)

Comments and Suggestions for Authors

Paper has not improved over multiple revisions, not interested, reviewing it anymore

Comments on the Quality of English Language

Poor

This manuscript is a resubmission of an earlier submission. The following is a list of the peer review reports and author responses from that submission.

Round 1

Reviewer 1 Report

Comments and Suggestions for Authors

The study presents a novel and potentially impactful approach to the grading of ccRCC using machine learning and radiomics.

The idea is good, and the structure is well-organized. However, there is a couple of concerns to be addressed. 

The use of PyRadiomics for feature extraction and the subsequent standardization of these features are well-executed. However, the high number of features (approximately 1500 per image) might introduce complexity and potential overfitting issues in the model. How do the author reduce the issue of overfitting. 

Please provide details how they optimize the model details.

Comments on the Quality of English Language

NA

Author Response

N/A

Reviewer 2 Report

Comments and Suggestions for Authors

Summary: The study has used multiple cohorts to predict pathologic grade of kidney cancers from CT scans. Models perform relatively well at their designated task. Overall, the manuscript does not provide sufficient rationale for the study, and many elements that could be considered novel are not properly explained. This combined with grammatical errors and extensively repeated tables lead to an underwhelming paper that requires further development before publication.

Abstract

 Obvious grammatical errors

Introduction:

 Very detailed description of the tumor, its presentation and characteristics, but does not seem to have a connection to their study

 Sections of text are basically repeated verbatim

 This introduction is packed with a lot of not very useful information that seems to hide the fact that this is a very simple study. I do not know why a lot of this information is included, it does not seem to have any bearing on the study

Materials and Methods

 Large cohorts with ISUP grading, all contrast enhanced

 Why were patients with multiple tumors excluded? Why were patients with bilateral tumors excluded? No explanation is given

 Good description of the various devices used to acquire the images

 No description of what qualifies a “Low quality” CT scan. This is important exclusion criteria that is not explained

 Good explanation of why grades 1+2 are grouped and 3+4 are grouped. However, why not just do a 4 class model? Why binarize the output?

 Strange arbitrary smoothing operation applied in the pipeline

 Segmentations on cohort 2 were very different then cohort 1. Did this affect results?

 Still unclear why we need to separate the tumor into exterior, interior, and core regions. It seems they extracted radiomics features from these different regions and then trained different models. Why?

Results:

 Repeats the same type of table 10 times, which basically repeats the same information for the different cohorts.

 Very little intuition regarding the rationale or what exactly was attempted in what fashion and what the takeaways are.

Comments on the Quality of English Language

Very poor, numerous grammatical and spelling errors.

Author Response

N/A

Round 2

Reviewer 2 Report

Comments and Suggestions for Authors

This manuscript is not significantly changed from last time. Poorly written and significantly flawed.

Comments on the Quality of English Language

Very poor